# Effectiveness of the “Planning Health in School” Programme on Children’s Nutritional Status

**DOI:** 10.3390/ijerph182312846

**Published:** 2021-12-06

**Authors:** Margarida Vieira, Andreia Teixeira, Graça S. Carvalho

**Affiliations:** 1Research Centre on Child Studies, University of Minho, 4710-057 Braga, Portugal; graca@ie.uminho.pt; 2Department of Community Medicine, Information and Decision in Health, Faculty of Medicine, University of Porto, 4200-319 Porto, Portugal; andreiasofiat@med.up.pt; 3Center for Health Technology and Services Research, Faculty of Medicine, University of Porto, 4200-319 Porto, Portugal; 4ADiT-LAB, Instituto Politécnico de Viana do Castelo, Rua Escola Industrial e Comercial Nun’Álvares, 4900-347 Viana do Castelo, Portugal

**Keywords:** obesity prevention, school-based intervention, nutritional status, eating behaviour, physical activity

## Abstract

Effective interventions for guiding children to change behaviours are needed to tackle obesity. We evaluated the effectiveness of the ‘Planning Health in School’ programme (PHS-pro) on children’s nutritional status. A non-randomised control group pretest-posttest trial was conducted at elementary schools of a sub-urban municipality in Porto’s metropolitan area (Portugal). A total of 504 children of grade-6, aged 10–14, were assigned in two groups: children of one school as the intervention group (IG), and three schools as the control group (CG). Anthropometric measures included height, weight, waist circumference (WC), BMI and waist-to-height ratio (WHtR), and lifestyle behaviours (self-reported questionnaire) were assessed at baseline and after PHS-pro. IG children grew significantly taller more than CG ones (*p* < 0.001). WC had reduced significantly in IG (−0.4 cm) whereas in the CG had increased (+0.3 cm; *p* = 0.015), and WHtR of IG showed a significant reduction (*p* = 0.002) compared with CG. After PHS-pro, IG children consumed significantly fewer soft drinks (*p* = 0.043) and ate more fruit and vegetables daily than CG. Physical activity time increased significantly in IG (*p* = 0.022), while CG maintained the same activity level. The PHS-pro did improve anthropometric outcomes effectively leading to better nutritional status and appears to be promising in reducing overweight and obesity.

## 1. Introduction

Overweight and obesity have spread worldwide very fast in the last thirty years, with children most affected [1]. The prevalence of this serious problem in children has already exceeded the 20% in all Europe [2].

According to the last update of the World Obesity Federation [3], more than 30% of Portuguese children are overweight or obese, with girls and boys of 11-years old occupying the seventh position of all-European countries [4]. Two previous studies have found these results in Portuguese children and adolescents [5,6].

Obesity is no longer seen as the simple result of a daily and chronic imbalance between consumed and expended calories but instead recognised as a complex problem with multiple etiological overlapping factors. Environmental factors, such as unhealthy eating and inactivity, inappropriate food systems and transportation resources, are the major drivers to explain the dimension of obesity [7]. Children live with excessive availability of high energy-dense foods complemented by a lack of energy expenditure [8,9,10].

The consequences of being overweight and obese during growth are well documented, showing a general risk in physical and psychological health. Obese children are more likely to experience mental health problems such as depression or anxiety, low self-esteem, bullying and behaviour problems than non-obese, and also exhibit a higher frequency of cardiovascular risk factors like high blood pressure, dyslipidaemia, hypertension, insulin resistance plus other clinical consequences as asthma, type I diabetes and abnormalities of foot structure [11,12]. Furthermore, a large body of evidence indicates that obesity in youth imposes an even larger burden in adulthood with a significantly increased risk of premature mortality with substantial and adverse long-term consequences as cancer, cardiovascular diseases and diabetes [13].

Considering the current knowledge and the delay in tackling this epidemic, effective interventions should be the focus, bringing new perspectives and solutions for guiding children to healthier habits. Significant evidence indicates school-based intervention programmes as the most appropriate approach to support young people in the promotion of healthy habits to reduce rates of overweight and obesity [14,15,16].

In this perspective, a school-based intervention called the “Planning Health in School” programme (PHS-pro) was designed and implemented based on theoretical constructs of the Transtheoretical Model (TTM) and stages of behaviour change so children could progress through the five stages developed by Prochaska and colleagues [17].

We hypothesised that the implementation of PHS-pro, a behavioural change intervention about healthy eating and active living issues in grade 6 children, over an academic year, would lead to positive changes in their anthropometric measures towards their healthy nutritional status. The primary aim was to determine anthropometric changes in children following the PHS-pro as compared to a control group.

## 2. Materials and Methods

The PHS-pro aimed to promote healthy eating and active living in children and was implemented over one school year. The PHS-pro priorities were set based on international guidelines [18]: (i) adequate consumption of fruit and vegetables (F&V)—five servings/day; (ii) decreasing high sugar food and beverage intake (iii) decreasing high fat and energy-dense food consumption; (iv) one hour of physical activity daily and TV watching below two hours/day.

The PHS-pro was designed to produce a strong cognitive, attitudinal, and behavioural impact rather than only build upon limited pieces of advice like “eat less, move more” [19]. Thus, the intervention was based upon the TTM, supporting behaviours change as a process in which children progress through five stages, moving from inaction to maintaining healthy behaviours. In addition, the core constructs of TTM (processes of change, decision balance, self-efficacy) were also applied for supporting their progress across stages and to engage them to participate actively in order to increase knowledge and nutritional literacy, to develop competencies for decision-making on healthy choices [17].

The educational components of the intervention consisted of a set of eight learning modules and participatory activities, following the five stages of TTM and the participatory methodology principles. Accordingly, children selected eight topics about nutrition and physical activity that were implemented monthly in the Natural Science classes, delivered by a nutritionist and supported by the teacher. The process of behaviour change was monitored through food records [20]. Moreover, parental involvement was required in two meetings for encouraging parents to increase F&V availability at home and support children’s behaviour changes. Likewise, the availability of F&V in the school canteen was ensured and assessed periodically over the intervention.

### 2.1. Participants

A non-randomised control group pretest-posttest trial was conducted between September 2011 and May 2012 at the four existing elementary schools of a municipality (Trofa) integrated into the second-largest metropolitan area of Portugal—Porto.

Schools and participants were assessed on several environmental factors to ensure equivalence and organise two balanced samples: school grade, nutrition and health-related school curriculum, textbooks, physical education classes and school food services.

A total of 504 children were registered in grade 6: 47.6% of children attended one school and was selected for the intervention group (IG), while the other three schools served as the control group (CG). No chance of contamination between groups was possible. Random assignment of children among different classes and schools to be the CG or IG was impossible due to practical reasons of working, and it would generate bias in outcome results. Indeed, synergies should be created over daily contacts between subjects of IG for a successful implementation of the programme in a school environment [21].

The study protocol was approved by the Scientific Council of the Institute of Education of the University of Minho and ethical permission was obtained from each Pedagogical School Board.

Parents were invited for a school meeting and were fully informed about the study procedures, risks and benefits of participation, confidentiality and signed an informed consent. A similar procedure was delivered to children on the first day of school.

Of the 504 children recruited, 26 (5.2%) were excluded: one refused to participate, four moved to another school, and 21 had special education needs (see Figure 1).

Participants’ inclusion criteria were the following: complete data of baseline and follow-up assessments, not receiving any nutritional intervention outside the school, taking no medication that might interfere with body metabolism or be associated with weight change. The final sample was composed of 449 children: 219 in IG and 230 in CG.

### 2.2. Assessments

Two assessments were carried out: baseline data collection occurred in the beginning of the school year (September 2011) and follow-up data collection occurred promptly after the 8 months school intervention (between May and June 2012). Each assessment included anthropometric measures for nutritional status, and eating, physical, and sedentary behaviours using a self-reporting questionnaire.

#### 2.2.1. Anthropometric Measures

Height, weight and waist circumference measurements were collected during physical education classes (shorts, t-shirt without footwear), using standard procedures [22]. Height was measured to the nearest 0.1 cm using a portable stadiometer (Seca model 214, Hamburg, Germany). Weight was recorded to the nearest 0.1 kg using a portable digital scale (Seca model robusta 813). Waist circumference (WC) was assessed in the anatomical landmark midpoint between the lowest rib and the iliac crest at the end of expiration, and was taken three times to improve precision [23].

To ensure measurement reliability, the equipment was calibrated before each day of assessment. All assessments were obtained by one examiner (nutritionist and research coordinator) assisted by a teacher nominated by the Health Promoter Office of schools for recording the collected data.

Body Mass Index (BMI) values for normal weight, thinness, overweight and obesity were defined using cut-off points of Cole and collaborators [24,25]. WC was classified according to UK reference percentiles to gather an overview of adiposity due to its correlation with intra-abdominal fat mass [26], and waist-to-height ratio (WHtR) calculated, dividing WC by height [27]. Indeed, BMI used alone to access improvements on nutritional status is not reliable due to its low sensitivity for body fat percentage, whereas WC and WHtR can accurately identify fat tissue changes [28]. Height, weight and BMI for age and sex-specific z-scores were also obtained to give reliability for comparative analysis with international studies [29].

#### 2.2.2. Lifestyle Behaviours

A self-reporting questionnaire was used to collect eating habits, physical activity and sedentary behaviours. Participants were asked to report their usual food consumption weekly with a 58-item food frequency questionnaire (FFQ). The guidelines by Cade and colleagues [30] were followed to organise the FFQ and the food list adapted to suit to PHS-pro priorities. Answers were displayed on a 5-point ordinal scale, ranging from never (0) to every day (4), as used by Block [31].

Food frequency consumption data allowed evaluating children’s health-risk eating habits. Five food clusters were created according to PHS-pro priorities and a score system was introduced to estimate servings per day from the times-per-week system [32]. The five category options for answering in FFQ were recoded into outcome variables: (i) never = 0; (ii) once a week = 0.142; (iii) two to four times a week = 0.357; (iv) four to six times a week = 0.714; (v) everyday = 1. Thus, for each response, a score was assigned to give the estimation of servings/day.

Food items were grouped to provide central nutrients (vitamins, minerals and dietary fibers), fats or free sugars. Accordingly, five food clusters were created: (i) F&V including six items: fruits, vegetable soup at lunch, vegetable soup at dinner, salad or greenery at lunch, salad or greenery at dinner, fresh fruit juices; (ii) High Energy Dense food, including two-items: pastry (croissants, pancakes, donuts, cakes, pastries), cookies/biscuits; (iii) Soft drinks, including two-items: soda, added sugar squash juices; (iv) High Sugar Food, including six-items: chocolate, candies, gums, ice-creams, marmalade (jelly, jams), sugar; (v) High fat food, including three-items: homemade French fries, chips, rissoles (pasties, croquettes).

Physical activity was assessed using four items of participation in organised sports activities out of school and the number of hours/week. Response options ranged from “never” to “7 or more hours” in a total of six categories. The school curriculum included two physical education classes with a total of 135 min/week, which gives a mean of 19.28 min daily.

Sedentary behaviours were assessed with two items asking the number of hours spent watching TV and playing video games daily. To compare time spent in physical activity recommendations with sedentary behaviours, children answers were calculated in minutes/day.

The questionnaire was pilot tested to an eligible sample of thirty children from the same region, grade and age, but not involved in the study.

### 2.3. Statistical Analysis

The normality of the data was verified with Q-Q plots. Unpaired or paired two-sided t-test was used for comparison normally distributed data and were characterised by means and standard deviations (SD). In cases of non-normally distributed variables, Mann-Whitney or Wilcoxon tests were applied, and data were characterised by medians (Q2; 50th percentile) and interquartile ranges (IQR: (Q1–Q3); 25–75th percentile) or 5th and 95th percentiles (P5; P95) when differences were not visible with IQR. Chi-square test and Fisher’s test were used to compare categorical variables, which were characterised by absolute (n) and relative (%) frequencies.

Changes in anthropometric variables were computed as the difference between measures at baseline and follow-up. The intervention effect identified as Delta (Δ) is the change difference of anthropometric variables between groups. A multiple linear regression analysis was applied to adjust significantly different variables. Values of *p* < 0.05 were considered significant. SPSS statistical software package version 22.0 was used for data analysis.

## 3. Results

Demographical and socio-economical characteristics of IG and CG children (Table 1) showed no significant differences at baseline regarding age, gender, and maternal education level. However, significant differences between groups were found regarding paternal education level (*p* = 0.017), maternal and paternal occupation (*p* = 0.020 and *p* = 0.013, respectively).

Statistical analysis of IG and CG child anthropometric data at baseline (Table 2) showed no significant differences, except for weight: the CG was significantly heavier (*p* = 0.012), and consequently, the BMI was higher (*p =* 0.017). Also, CG overweight (33.0%) and obese (15.2%) children were in significantly higher percentage (*p* = 0.038) as compared to IG children, 27.4% and 10.5%, respectively. Furthermore, a clear majority of the studied sample was found to be above the 90th percentile of WC by age and sex (61.9%).

### 3.1. Changes in Anthropometric Measures

As expected, both groups grew in height and weight over the study period (10 months); however, significant differences between groups were found. Table 3 shows the changes in anthropometric measures by group and gender following the intervention. Regarding the height, significant differences between groups were found (*p* < 0.001). The IG grew significantly more than the CG among girls (+0.4 cm; *p* = 0.011) and boys (+0.5 cm; *p* = 0.003). Although the CG was heavier at baseline, both groups gained almost the same weight (IG: 2.5 kg; CG: 2.4 kg). Naturally, the BMI increased between baseline and follow-up in both groups; however, CG (0.2 kg/m^2^) almost doubled compared to IG (0.1 kg/m^2^).

Significant differences were also found in WC (*p* = 0.015): IG children showed a decrease while CG had an increase. Gender effects in WC values were also found among girls (IG: −0.5 cm and CG: +0.4 cm; *p* = 0.023). Considering that, girls of CG increased more than boys (+0.4 v. +0.2 cm, respectively), whereas girls of IG reduced more than boys (−0.5 v. −0.3 cm). WHtR and z-scores variables exhibited the same statistical behaviour as their matching variables.

Given these results, and considering that IG and CG children showed a significant difference in weight at baseline, we carried out a multiple linear regression analysis to adjust the weight variable and determine whether the significant differences in height, WC and WHtR have remained. After adjustment, the significant differences found in height, WC and WHtR between the two groups persist.

### 3.2. Changes in Eating Habits and Physical Activity

Differences between groups on eating habits changes, physical activity and sedentary habits over the intervention are shown in Table 4.

Regarding eating habits, the five food clusters represent the consumptions of F&V, high-energy dense food, soft drinks, high sugar food and high-fat food. Daily consumption of F&V showed a negative change in CG between baseline and follow-up, with the median value decreasing significantly from 3.2 to 3.0 serving/day (*p* = 0.002); in contrast, no statistical change was found in IG, but a tendency to increase fruit intake with an improvement of 2.8 to 3.2 servings/day. Regarding soft drinks consumption, IG children reported consuming significantly less, as the median decreased from 0.7 to 0.5 servings/day (*p* = 0.043) while CG kept the median 0.5 (*p* = 0.097). No evidence was found of the PHS-pro effect on the consumption of high sugar, high-fat and high-energy dense snacks.

Regarding physical activity, baseline data showed half of the children of both groups with a physical activity far from the recommended 60-min/day (IG: 23.58 and CG: 19.28 min/day). However, after the intervention, the IG significantly increased daily minutes (more 4.3 min/day) spent on physical activities (*p* = 0.022), while CG children maintained the same level of physical activity, except 5% of them that increased the time of physical activity (Baseline P95: 62.18 and Follow-up P95: 79.28 min/day; *p* = 0.019). For the time spent watching TV and playing video games, no significant differences between groups were found either at baseline or follow-up. However, 5% of the IG children reported significantly more time spent with these activities (*p* = 0.006).

## 4. Discussion

In this study, nearly one-third (30.3%) of children had a BMI above 25 kg/m^2^ and about one-eighth (12.9%) above 30 kg/m^2^ being classified as overweight and obese. Additionally, a clear majority of children were above the 90th WC percentile, a cut-off value linked to high abdominal fat mass and cardiometabolic risk [23,33]. Furthermore, the high prevalence of overweight and obesity was surpassed by a high prevalence of children above the 90th WC, supporting the urgent need for intervention. These results are in line with other previous Portuguese studies [5,6,34].

A variety of different indicators are used to measure socioeconomic status in study interventions targeting children such as parent’s education, occupation, income, or social class [35]. There is evidence of a significant relationship between parents’ years of education and childhood obesity, being parental education the most important variable among the many ascertained socioeconomic status indicators, since it showed more significantly associated with measures of the nutritional status and health-related behaviours of the children than other social variables [36,37]: a higher socioeconomic status is associated with lower adiposity, and children from low-income families are more likely to have obesity or overweight, particularly in Western Europe [38].

In this study, we accessed parent’s education and occupation to define the socioeconomic status of children. At baseline, there was no significant differences regarding maternal education level. However, there were statistical differences between groups regarding paternal education level and maternal and paternal occupation, beneficiating the CG compared with the IG: more fathers had a university degree in the CG (8.3% vs. 5%), and more fathers with grade 6 school (39.6% vs. 30.1%), which was the most prevalent grade among the two studied samples; with respect to maternal and paternal occupation a higher percentage of mothers who had a job (81.3% vs. 76.7%), as well as a higher percentage of fathers (90.9% vs. 84%), which translates into earning a greater income for the families of the CG children. Although the socioeconomic status of CG may show a greater degree compared with the IG, CG children were significantly heavier, with more of them classified with overweight (33% vs. 27.4%) and obesity (15.2% vs. 10.5%).

Several studies have identified healthy eating and lifestyle behaviours as the best solution to revert obesity [7,39]. Also, some school-based interventions have given clear evidence of being successful in obesity control [14,16].

The results of this study showed favourable changes in children’s nutritional status participating in the PHS-pro. Boys and girls of IG had a significant height increase and WC reduction; in contrast, CG grew less and increased the WC. In childhood, healthy development is characterised by a gradual increase of height and weight, which influences consistently other anthropometric measures, such as the WC. However, the WC can be a marker of adiposity-related morbidity, and the increase of WC in these ages may not be considered a healthy outcome [28,40]. There has been a few research investigating WC changes in children; the majority of school interventions use BMI as the principal obesity indicator. However, our WC results are consistent with the small amount of data that does exist [39,41,42].

Additionally, the PHS-pro showed to be effective in improving daily F&V intake, reducing soft drink consumption, and encouraging the dedication of more time to physical activities. Conversely, CG children did not show any improvements regarding these habits; indeed, worse behavioural measures were found: lower daily F&V consumption, less time spent in physical activity and more time in front of screens. Comparable findings to this study were found in a two-year school programme focused on increasing F&V, reducing sweetened drink intake and improving WC [43] and with two other programmes [42,44].

Baseline consumptions of F&V were below the recommendation of 5 servings/day, with half of IG children consuming 2.8 and CG 3.2 servings. Favourable changes were observed on the daily consumption of F&V in IG children at the end of the intervention but not in CG. A study held in 33 countries examining F&V consumption among children showed that most children do not consume F&V daily, and a significant decrease in fruit consumption has occurred in five countries where Portugal was included [45]. An analysis of seven school-based nutrition programmes seeking to increase F&V consumption has shown weaker results [46] than those of this study. Ultimately, a recent literature review to report the effectiveness of dietary interventions in childhood obesity published between 2009 and 2021, discloses that a large number of studies, mainly based on school interventions, did not show any results or did not obtain effective results [47].

In high-sugar, high-fat and high-energy dense snacks, no evidence of the PHS-pro effect on self-reported consumption were found. A possible explanation could be that the FFQ is not sensitive enough to detect changes, although it was previously validated for the target population. In future interventions, brief-screening tools to assess specific eating behaviour (i.e., F&V or high-sugar foods) should be used to detect changes instead of FFQ. This issue was considered during the research design; however, PHS-pro was a pilot intervention, and a wider questionnaire should give larger perspectives. On the other hand, the eating behaviour of IG over the intervention period was tracked with food records [20].

Positive changes in the physical activity of IG children were found as compared to the CG. At baseline, time spent with physical activities was much below 60 min/day in both groups. In Portugal, school physical education ensures 135 min/week, which gives 19.3 min/day, far shorter than 60 min/day recommended for children [18]. Half of the children did not practice any sport or other extra-school physical activity. On the other hand, time spent with sedentary tasks (watching TV and playing computer games) was very high, with half of the children reporting 90 min daily on these activities. The PHS-pro dedicated one module to active living, although a significant increase of 4.30 min/day was obtained. A systematic review involving 14,326 children to access the effectiveness of interventions on physical activity showed an increase of 4.00 min/day [48], which is similar to our results. More recently, another review evaluating 13 studies to access school-based interventions for promoting physical activity, a significant increase was found in physical activity levels in five studies [16].

Educational interventions in the school context can successfully improve health status through daily behaviour changes, as this study showed. Nevertheless, the motivation process for healthy behaviours must be delivered regularly and continuously, allowing progress over the stages of change as TTM prescribes [17]. The promotion of healthy behaviours during childhood has to combine methodologies to encourage children’s participation and monitor their needs, as implemented in the PHS-pro. In contrast, sporadic actions to fulfil public policy schedules have shown to be ineffective [7,14,49].

This study has some limitations. The study design was not randomised, and there is always a risk of selection bias; however, the similarity of both groups was an important issue for comparison between children exposed (IG) and not exposed (CG) to the PHS-pro. Both groups were homogeneous regarding age and gender, had high participation rates and low dropouts. The eating habits data were self-reported by children who may not recall data correctly. Self-reports to evaluate eating and physical activity habits are commonly used to assess behavioural data of children; however, this kind of data is susceptible to report bias by inaccurate responses and errors in recall [50,51]. Therefore, the interpretation of the study outcomes must be treated with considerable caution. Further, the study was conducted in a small municipality with four elementary schools, and the results may not be generalisable to other populations; however, these results are useful to conduct larger studies.

The significant strength of this study is the applicability of the programme to the real world in a school context over an academic year. The PHS-pro implementation conciliated school schedules, curriculum contents and infrastructures, school staff and family efforts.

Across Europe and North America, childhood obesity has spread very fast in recent decades. Nevertheless, despite efforts to promote F&V intake and reduce unhealthy foods intake and physical inactivity among children, most of them do not meet recommendations. The current study contributes to the body of knowledge in the largely unexplored area of interventions for preventing obesity and therefore offers evidence to guide health promotion practice in schools. Also, findings of PHS-pro can be relevant for replicating in larger studies in other regions and other group ages.

The strong argument for health interventions in schools is that children can be reached there with relatively less effort and ensured global coverage. Furthermore, the school setting provides all conditions to successfully initiate guidance on children’s behaviour for good life habits over the academic years. Our findings support the success of the PHS-pro educational model planned according to the TTM, helping children to adopt a healthful diet, specifically by encouraging them to F&V intake, to reduce soft drinks intake, and to be more active physically. In fact, to improve the likelihood of effectiveness, the interventions need to be based on behaviour change theories that focus on promoting action. The PHS-pro was designed under the TTM of behaviour change, allowing to follow-up children in their process of change. Therefore, we strongly suggest that the implementation of educational interventions based on TTM of stage change should be encouraged. The positive changes found in eating habits and physical activity resulted in effective changes in their body composition, particularly the WC reduction that brings forward fat tissue changes. Children’s nutritional status improvements lead to better and healthy development, being a strategy of foremost importance for reducing overweight over childhood and preventing obesity.

## 5. Conclusions

This study found that the PHS-pro did improve anthropometric outcomes effectively, leading to better children’s nutritional status, and showed to be effective in reducing soft drink consumption, improving daily F&V intake, and encouraging the dedication of more time to physical activities. Thus, this study provides evidence to suggest that the PHS-pro has the potential to guide children for healthy habits and improve key obesogenic behaviours, the major requirement to prevent children’s overweight.

## Figures and Tables

**Figure 1 ijerph-18-12846-f001:**
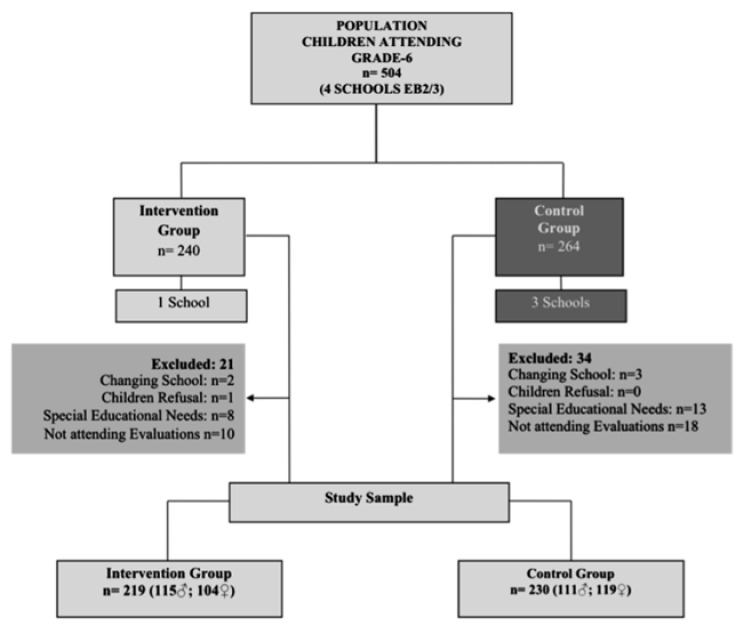
Flow chart of recruitment and sample selection.

**Table 1 ijerph-18-12846-t001:** Demographical and socio-economical characteristics of intervention group (IG) and control group (CG).

	Total		IG		CG		*p*-Value
	*n* = 449		*n* = 219		*n* = 230		
Age (years), mean (SD)	11.2	0.6	11.2	0.6	11.2	0.64	0.736 ^a^
Gender, n (%)							0.368 ^b^
Boys	226	50.3	115	52.5	111	48.3	
Girls	223	49.7	104	47.5	119	51.7	
Maternal Education, n (%)							0.058 ^a^
Do not know	27	6	16	7.3	11	4.8	
Primary school (grade 4)	91	20.3	40	18.3	51	22.2	
Secondary school (grade 6)	114	25.4	46	21	68	29.6	
(grade 9)	108	24.1	52	23.7	56	24.3	
(grade 12)	82	18.3	50	22.8	32	13.9	
University degree	27	6	15	6.8	12	5.2	
Paternal Education, n (%)							0.017 ^a,^*
Do not know	45	10	28	12.8	18	7.8	
Primary school (grade 4)	86	19.2	37	16.9	49	21.3	
Secondary school (grade 6)	157	35	66	30.1	91	39.6	
(grade 9)	78	17.4	47	21.5	31	13.5	
(grade 12)	52	11.6	30	13.7	22	9.6	
University degree	30	6.7	11	5	19	8.3	
Maternal Occupation, n (%)							0.020 ^b,^*
No answer	11	2.4	10	4.6	1	0.4	
Unemployed	80	17.8	39	17.8	41	17.8	
Worker	355	79.1	168	76.7	187	81.3	
Retired	3	0.7	2	0.9	1	0.4	
Paternal Occupation, n (%)							0.013 ^b,^*
No answer	18	4	13	5.9	5	2.2	
Unemployed	35	7.8	22	10	13	5.7	
Worker	393	87.5	184	84	209	90.9	
Retired	3	0.7	0	0	3	1.3	

^a^, χ^2^ test; ^b^, Fisher’s exact test. * *p* < 0.05.

**Table 2 ijerph-18-12846-t002:** Anthropometric measures of intervention group (IG) and control group (CG) at baseline.

	Total	IG	CG	*p*-Value
	*n* = 449	*n* = 219	*n* = 230	
Anthropometric data, Mean (SD)				
Weight (kg)	46.1 (1.1)	44.7 (11.1)	47.4 (11.5)	0.012 *
Weight z-score	1.0 (1.1)	0.9 (1.1)	1.2 (1.1)	0.005 *
Height (cm)	148.8 (7.1)	148.3 (7.0)	149.3 (7.2)	0.11
Height z-score	0.6 (1.0)	0.5 (1.0)	0.7 (1.0)	0.066
Waist Circumference	74.8 (1.1)	73.8 (10.8)	75.7 (10.9)	0.065
BMI (kg/m^2^)	20.6 (4.0)	20.2 (4.0)	21.1 (4.1)	0.017 *
BMI z-score	1.0 (1.2)	0.8 (1.3)	1.1 (1.2)	0.008 *
WHtR (waist-to-height ratio)	0.5 (0.1)	0.5 (0.1)	0.5 (0.1)	0.135
IOTF Classification, n (%)				0.038 ^b,^*
Underweight	16 (3.6)	12 (5.5)	4 (1.7)	
Normal	239 (53.2)	124 (56.6)	115 (50.0)	
Overweight	136 (30.3)	60 (27.4)	76 (33.0)	
Obesity	58 (12.9)	23 (10.5)	35 (15.2)	
WC Percentiles, n (%)				0.106 ^b^
P5	0 (0.0)	0 (0.0)	0 (0.0)	
P10	3 (0.7)	3 (1.4)	0 (0.0)	
P25	12 (2.7)	9 (4.1)	3 (1.3)	
P50	44 (9.8)	23 (10.5)	21 (9.1)	
P75	64 (14.3)	31 (14.2)	33 (14.3)	
P90	48 (10.7)	27 (12.3)	21 (9.1)	
P95	278 (61.9)	126 (57.5)	152 (66.1)	

^b^ χ^2^ test. * *p <* 0.05.

**Table 3 ijerph-18-12846-t003:** Changes in anthropometric measures for intervention (*n* = 219) and control group (*n* = 230).

	IG ∆		CG ∆		*p*-Value
	Mean	SD	Mean	SD	
Height (cm)	3.5944	1.3127	3.0914	1.3055	<0.001 ^a,^*
Girls	3.4018	1.2548	2.9634	1.2789	0.011 ^a,^*
Boys	3.7685	1.3447	3.2286	1.3255	0.003 ^a,^*
Height z-score	0.0449	0.2382	−0.0212	0.2445	0.004 ^a,^*
Weight (kg)	2.4493	2.1711	2.4078	2.3756	0.847 ^a^
Girls	2.6058	2.1822	2.5966	2.3211	0.976 ^a^
Boys	2.3078	2.1608	2.2054	2.4269	0.738 ^a^
Weight z-score	−0.0189	0.2441	−0.0316	0.2521	0.588 ^a^
BMI (kg/m^2^)	0.1216	0.9395	0.2078	1.0113	0.351 ^a^
Girls	0.2445	0.9379	0.3374	0.9398	0.462 ^a^
Boys	0.0105	0.9310	0.0689	1.0695	0.662 ^a^
BMI z-score	−0.0762	0.2971	−0.0632	0.3264	0.659 ^a^
Waist Circumference (cm)	−0.3745	2.8105	0.2970	2.9844	0.015 ^a,^*
Girls	−0.4708	3.0458	0.4297	2.8240	0.023 ^a,^*
Boys	−0.2875	2.5898	0.1546	3.1539	0.250 ^a^
WHtR (waist-to-height ratio)	−0.0142	0.0195	−0.0082	0.0204	0.002 ^a,^*

IG, intervention group; CG, control group. (Δ) indicates the change observed for the specific variable over the intervention period. ^a^: *t*-test. *: *p ≤* 0.05. Table 3 is presented with more than two decimal places in order to allow to look at the differences found.

**Table 4 ijerph-18-12846-t004:** Changes on eating, physical activity, and sedentary habits for intervention (*n* = 219) and control group (*n* = 230).

	Baseline			Follow-Up			
	IG Median [IQR] or (P5;P95)	CG Median [IQR] or (P5;P95)	*p*	IG Median [IQR] or (P5;P95)	CG Median [IQR] or (P5;P95)	*p*	*p*-Value of Change
Eating habits (servings/day)							
F&V	*n* = 206 2.83 [1.71–4.37]	*n* = 217 3.19 [2.26–4.35]	0.167 ^b^	*n* = 207 3.19 [2.05–4.42]	*n* = 219 3.00 [1.77–4.12]	0.158 ^b^	IG:0.068 ^a^ CG:0.002 ^a,^***
Soft drinks	*n* = 214 0.70 [0.28–1.14]	*n* = 226 0.49 [0.28–1.06]	0.072 ^b^	*n* = 213 0.49 [0.28–1.06]	*n* = 220 0.49 [0.28–1.06]	0.132 ^b^	IG: 0.043 ^a,^*** CG:0.097 ^a^
High sugar food	*n* = 214 0.84 [0.42–1.55]	*n* = 226 0.91 [0.42–1.49]	0.944 ^b^	*n* = 212 0.91 [0.44–1.41]	*n* = 225 0.91 [0.42–1.47]	0.636^b^	IG: 0.591 ^a^ CG: 0.401 ^a^
High-fat food	*n* = 216 0.49 [0.28–0.99]	*n* = 221 0.49 [0.28–0.84]	0.573 ^b^	*n* = 212 0.49 [0.28–0.84]	*n* = 224 0.49 [0.28–0.84]	0.541 ^b^	IG: 0.643 ^a^ CG: 0.615 ^a^
High-energy dense food	*n* = 212 0.70 [0.35–1.06]	*n* = 226 0.49 [0.35–1.00]	0.777 ^b^	*n* = 219 0.70 [0.35–1.06]	*n* = 225 0.49 [0.28–1.00]	0.090 ^b^	IG: 0.253 ^a^ CG: 0.478 ^a^
Physical activities, (min/day)	*n* = 207 23.58 (19.28;72.44)	*n* = 221 19.28 (19.28;62.18)	0.171 ^b^	*n* = 217 27.88 (19.28;62.18)	*n* = 222 19.28 (19.28;79.28)	0.104 ^b^	IG:0.022 ^a,^*** CG:0.019 ^a,^***
Watching TV plus video games, (min/day)	*n* = 211 90.0 (45.0;390.0)	*n* = 223 90.0 (45.0;450.0)	0.335 ^b^	*n* = 218 90.0 (45.0;450.0)	*n* = 223 90.0 (45.0;450.0)	0.877 ^b^	IG:0.006 ^a,^*** CG: 0.235 ^a^

IG, intervention group; CG, control group; IQR, interquartile range; P5, percentile 5; P95, percentile 95. ^a^: Wilcoxon test. ^b^: Mann-Whitney test. *: *p* ≤ 0.05.

## Data Availability

The data presented in this study are available on request from the corresponding author. The data are not publicly available due to data privacy of the participants.

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
