# Peer review of "Effectiveness of the “Planning Health in School” Programme on Children’s Nutritional Status"

_ijerph, 2021, doi:10.3390/ijerph182312846_

Round 1

Reviewer 1 Report

This study shows a successful evidence of school-based interventions in healthy eating and lifestyle behavior: It shows favorable and in my opinion important changes in children’s nutritional status (more F and V daily consumption and physical activity) and in some anthropometric measures.

More programs/interventions, as you conducted, in healthy eating and lifestyle behavior in school-aged children are needed.

RESULTS:

Page 5 line 209-2010: I Have a questions about this sentence and with results shown in Table 2: “Furthermore, a clear majority of the studied sample was found to be above the 90th percentile of WC by age and sex (61.9%)” When I read this sentence I understand that above 90th includes percentiles 90th and 95th.

In table 2 this % includes p95? Is this sentence ok?

Author Response

We sincerely thank the reviewers for careful reading of our manuscript and the constructive comments and suggestions. Our response follows the reviewer comments.

Reviewer 1

This study shows a successful evidence of school-based interventions in healthy eating and lifestyle behavior: It shows favorable and in my opinion important changes in children’s nutritional status (more F and V daily consumption and physical activity) and in some anthropometric measures.

More programs/interventions, as you conducted, in healthy eating and lifestyle behavior in school-aged children are needed.

RESULTS:

Page 5 line 209-2010: I Have a questions about this sentence and with results shown in Table 2: “Furthermore, a clear majority of the studied sample was found to be above the 90th percentile of WC by age and sex (61.9%)” When I read this sentence I understand that above 90th includes percentiles 90th and 95th.

In table 2 this % includes p95? Is this sentence ok?

Thank you for your comment.

Yes, in our perspective the sentence is right.

The clear majority (61,9% of all children: IG+CG) is in fact children with P95. Children with P90 are not included.

We wanted to highlight the big number percentage of children above the 90th.

This is because children with a waist circumference greater than the 90th percentile are more likely to have multiple risk factors than children with a waist circumference that is less than or equal to the 90th percentile. Thus, the percentage of children with P90 included a <P90 or equal, and I did not included the sum of P90 and P95.

Reviewer 2 Report

This is an analysis of real life-setting educational intervention programme in children to prevent obesity. The study is very well designed, performed and presented. The introduction very well presents up- dated knowledge on childhood obesity. Methodology is properly described. 

However I would like the authors to comment on the differences between  CG and IG in terms of parental education and occupation, as well as anthropometric measures. How can you explain it and how it would influence the results? Can you comment what factors may modify the results?

Author Response

We sincerely thank the reviewers for careful reading of our manuscript and the constructive comments and suggestions. Our response follows the reviewer comments.

Reviewer 2

This is an analysis of real life-setting educational intervention programme in children to prevent obesity. The study is very well designed, performed and presented. The introduction very well presents up- dated knowledge on childhood obesity. Methodology is properly described. 

However I would like the authors to comment on the differences between  CG and IG in terms of parental education and occupation, as well as anthropometric measures. How can you explain it and how it would influence the results? Can you comment what factors may modify the results?

We would like to thank you for your valuable comments concerning our manuscript.
The socioeconomic characteristics of children of both groups were gathered to understand whether the two groups were homogeneous at baseline. In this study, we not investigated the association of the socioeconomic conditions of children with the prevalence of obesity found. The main purpose of the study was to find which results we could have with the change behaviour intervention programme in the intervention arm. Following the suggestions, we have added more explanations, which we hope meet with your approval, and now it is described in the discussion section.
